# Psychosocial and pandemic-related circumstances of suicide deaths in 2020: Evidence from the National Violent Death Reporting System

**Briana Mezuk**[1,2]*, **Viktoryia Kalesnikava**[1], **Aparna Ananthasubramaniam**[3], **Annalise Lane**[1], **Alejandro Rodriguez-Putnam**[1], **Lily Johns**[1], **Courtney Bagge**[4], **Sarah Burgard**[5,6], **Kara Zivin**[4,7]

**1** Department of Epidemiology, University of Michigan School of Public Health, Ann Arbor, MI, United States of America, **2** Institute for Social Research, Research Center for Group Dynamics, University of Michigan, Ann Arbor, MI, United States of America, **3** School of Information, University of Michigan School of Information, Ann Arbor, MI, United States of America, **4** Department of Psychiatry, University of Michigan School of Medicine, Ann Arbor, MI, United States of America, **5** Institute for Social Research, Population Studies Center, University of Michigan, Ann Arbor, MI, United States of America, **6** Department of Sociology, University of Michigan, Ann Arbor, MI, United States of America, **7** Department of Obstetrics and Gynecology, University of Michigan School of Medicine, Ann Arbor, MI, United States of America

* bmezuk@umich.edu

## Abstract

### Purpose

To describe and explore variation in 'pandemic-related circumstances' among suicide decedents during the first year of the COVID-19 pandemic.

### Methods

We identified pandemic-related circumstances using decedents' text narratives in the 2020 National Violent Death Reporting System. We use time-series analysis to compare other psychosocial characteristics (e.g., mental health history, interpersonal difficulties, financial strain) of decedents pre-pandemic (2017/2018: n = 56,968 suicide and n = 7,551 undetermined deaths) to those in 2020 (n = 31,887 suicide and n = 4,100 undetermined). We characterize common themes in the narratives with pandemic-related circumstances using topic modeling, and explore variation in topics by age and other psychosocial circumstances.

### Results

In 2020, n = 2,502 (6.98%) narratives described pandemic-related circumstances. Compared to other deaths in 2020 and to the pre-pandemic period, decedents with pandemic-related circumstances were older and more highly educated. Common themes of pandemic-related circumstances narratives included: concerns about shutdown restrictions, financial losses, and infection risk. Relative to decedents of the same age that did not have pandemic-related circumstances in 2020, those with pandemic-related circumstances were more likely to also have financial (e.g., for 25–44 years, 43% vs. 12%) and mental health

**Data Availability Statement:** The PI cannot share the data publicly because of the CDC restrictions on sharing Restricted Access Data as outlined in the

Data Use Agreement (Section IIc) the PI has with the NVDRS. The Restricted Access data includes the narrative texts used in this anlaysis. Any investigator who meets CDC eligibility critiera can request these data directly from the National Violent Death Reporting system (contact at nvdrs-rad@cdc.gov). Additional details for accessing these restricted access data are provided on the NVDRS website https://www.cdc.gov/nvdrs/about/nvdrs-data-access.html).

**Funding:** This project was supported by grants from the National Institute of Mental Health (R01-MH128198, to MPIs Mezuk and Zivin) and the American Foundation for Suicide Prevention (DIG-1-110-19, to Mezuk). The funders had no role in the conceptualization, analysis, interpretation, or decision to publish this manuscript.

**Competing interests:** The authors have declared tht no competing interests exist.

**Abbreviations:** CDC, Centers for Disease Control and Prevention; CME, Coroner/Medical Examiner; COVID, Coronavirus disease of 2019; LE, Law Enforcement; NVDRS, National Violent Death Reporting System; NVP, Negative predictive value; OR, Odds ratio; PPV, Positive predictive value; PrC, Pandemic-related circumstance.

(76% vs. 66%) psychosocial circumstances, but had similar or lower prevalence of substance abuse (47% vs. 49%) and interpersonal (40% vs. 42%) circumstances.

## Conclusions

While descriptive, these findings help contextualize suicide mortality during the acute phase of the COVID-19 pandemic and can inform mental health promotion efforts during similar public health emergencies.

## Introduction

Suicide is currently the 11th leading cause of death in the US, with over 46,000 deaths reported in 2022 [1]. Every year an additional 1.4 million Americans attempt suicide and nearly 10 million have serious thoughts about suicide [2]. For reasons that are not well-understood, suicide risk differs by age, gender, race, geography, and time of year; contextual factors such as social integration and economic uncertainty also contribute to suicide risk [1,3,4].

According to numerous surveys during the first year of the pandemic, Americans experienced heightened symptoms of emotional distress, depression, anxiety, and suicidality, and several reports expressed concerns that this would result in an increase in suicide mortality [5–10]. However, the CDC reported that the rate of suicide fell by 3% from 2019 to 2020 [11], continuing a brief declining trend that had begun pre-pandemic (rates declined 2.1% from 2018 to 2019) [12]. Notably, 2021 was marked by a substantial increase in suicide mortality of approximately 4% [13]. Whether, and in what ways, circumstances related to the pandemic contributed to suicide mortality in the acute phase of the crisis is not clear. Routine mortality data have only limited information on experiences prior to death and therefore and do not readily support investigations of the potential contribution of 'pandemic-related' factors [14].

The CDC launched the National Violent Death Reporting System (NVDRS) to address this data gap and provide information that can reveal interactions between individual and macro-correlates of suicide. The NVDRS is a state-based surveillance system that collates reports from coroner/medical examiners (CME), law enforcement (LE) officials, and vital statistics [15]. Each observation has qualitative text fields, called *case narratives*, which describe the contextual details of the incident as well as the events that preceded it, as that information is documented in the CME and LE reports. These narratives often contain information regarding circumstances in the decedent's life that are salient to their death including psychosocial factors such as recent difficulties in relationships, work, or school, as well as mental and physical health problems. In 2020, abstractors began including information in these narratives about pandemic-related circumstances (e.g., concerns about infection, job losses due to workplace closures) [16]. The contextual details of the case narratives make the NVDRS one of the only datasets that can examine correlates of suicide mortality in the context of COVID-19 pandemic on an individual level using data that is collected at the population scale [14,17].

We used these textual data to address three goals: (1) to identify pandemic-related circumstances (PrC) among suicide decedents in 2020, and compare the characteristics of decedents with PrC to other suicide deaths that occurred that year; (2) to describe the common themes in the narratives of decedents who had PrC using topic modeling; and (3) to examine the association between PrC with established psychosocial correlates of suicide (e.g., mental health, substance use, financial problems) in 2020. Collectively, we aim to provide a nuanced description of PrC among suicide deaths in the acute phase of the pandemic, with the goal of informing both research and practice on how to support public mental health efforts during periods of crisis.

## Materials and methods

### Data source

The NVDRS was implemented in 2003 with five participating states, and since then an increasing number of states contributed their annual data. At the time of analysis, the NVDRS included over 360,000 suicides and undetermined deaths from all 50 states (although not all states had 100% catchment at this time), District of Columbia, and Puerto Rico, from 2003 to 2020. Details of the data abstraction process appear elsewhere [15]. Briefly, CDC-trained abstractors compiled information from original source postmortem records, including death certificates, toxicology and autopsy results, and CME and LE reports, which they used to generate quantitative variables and write the CME/LE narratives [16].

This observational study uses NVDRS data for single suicides and deaths with undetermined intent from 2017 and 2018 (pre-pandemic period) and 2020 (first year of the pandemic). The decision to include undetermined deaths was based on prior studies showing that some manners of suicide (e.g., poisoning) are often misclassified as deaths with undetermined intent, especially for decedents with minority or low socioeconomic status [18,19]. Henceforth, we refer to all death cases as "suicides". The decision to exclude deaths that occurred as part of multiple victim incidents was based on prior studies showing that these suicides often have different contributing circumstances, such as higher rates of precipitating relationship problems [20].

We excluded data from 2019 (which were abstracted in 2020), because stay-at-home orders and other pandemic-related challenges (e.g., overwhelmed death investigation staff, staffing shortages in many state agencies) may have affected the accessibility and completeness of the narratives that year (per our email communications with NVDRS staff about data quality concerns). As detailed in **Fig 1**, we limited the sample to decedents aged ≥10 years that were coded as having "known" death circumstances, as instructed in the NVDRS Data User's Guide. For the 2020 data, 7,285 (20%) cases had no known circumstances. After applying this restriction, less than 1% of missingness was present in the NVDRS-coded circumstance variables.

For the time-series analysis comparing overall trends in 2020 to the pre-pandemic period, we restricted the sample to the 37 states that participated in 2017, 2018 and 2020 and to decedents with known date of death (n = 83,090). For the time-series analysis comparing PrC to non-PrC cases, we further restricted the sample to cases with narratives ≥35 characters long to ensure sufficient text for analysis (n = 83,056). Finally, for the comparison of PrC and non-PrC cases in 2020 only, including topic modeling of the PrC narratives, we used data from all states, decedents with "known" circumstances, and narratives ≥35 characters long (n = 35,861).

Access to restricted-access NVDRS data was approved by the CDC. The IRB at the University of Michigan determined that this study was exempt from human subjects regulations because the NVDRS data is limited to deceased individuals. Only de-identified data were released to the research team, and access to the 2020 NVDRS year of data was granted on October 19, 2022. In accordance with the requirements of the NVDRS data sharing agreement, all cited narratives have been modified to protect the privacy of decedents.

### Measures

**Decedent characteristics.** These quantitative variables included: age group [coded as 10–24, 25–44, 45–64, and 65 and older]; sex [male, female]; race/ethnicity [non-Hispanic white, American Indian/Alaska Native, Asian/Pacific Islander, Black/African American, Hispanic/

**Panel A.** Analytic sample comparing pandemic (year: 2020) to pre-pandemic (years 2017-2018) decedents (n=37 states that reported to the NVDRS during all years)

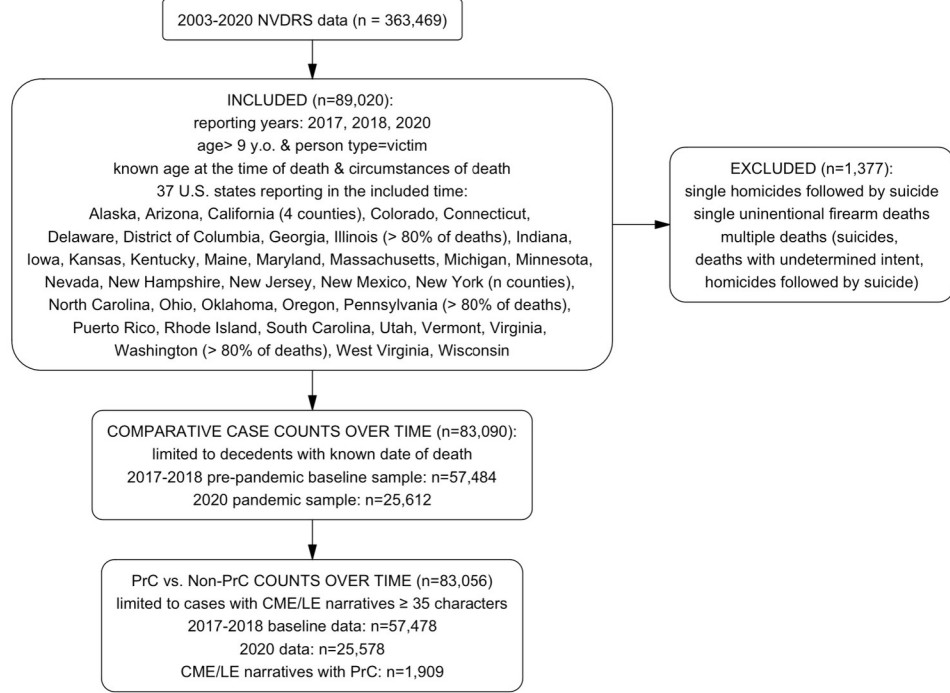

**Panel B.** Analytic sample for the year 2020 only (n=50 states that reported to NVDRS that year)

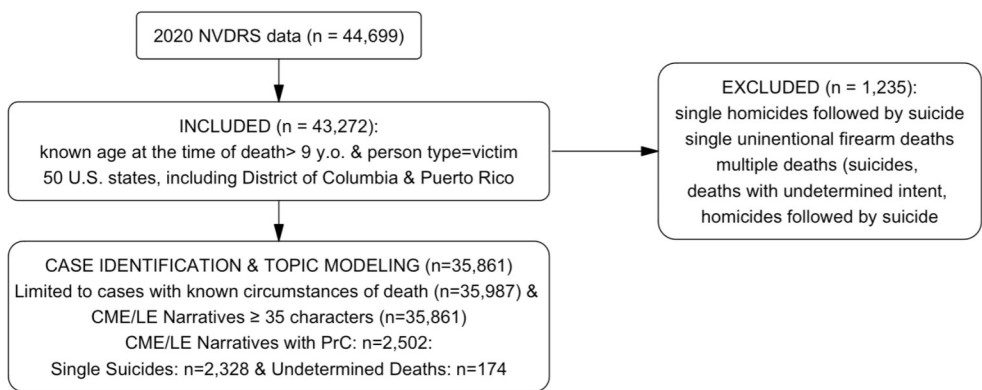

**Fig 1. Flowcharts illustrating the selection of the analytic samples.** The flowchart demonstrates the criteria used to generate different analytic samples used in this study. **Panel A** illustrates the process used to generate the analytic samples used for descriptive characteristics of decedents (years 2017–2018 and 2020) and temporal trends of suicide deaths in 2020 vs. pre-pandemic period. **Panel B** illustrates sample criteria for the analysis of 2020 used for identification of cases with PrC, association between PrC and other circumstances, and topic modeling analysis.

Latino, and Multiracial/Other]; education [≤8th grade, 9-12th grade, diploma/GED, some college, associate's degree, bachelor's degree, master's degree, doctorate degree]; relationship status [married/partnered, single/never married, and widowed/divorced/separated]; date of death (or, if unavailable, date of injury); autopsy status [full/partial vs. no autopsy]; and means of injury [firearm, poison, hanging/suffocation, sharp/blunt instrument, fall, drowning, vehicle, other]. Finally, because the existing NVDRS quantitative variables provide only limited detail regarding employment characteristics, we created a 6-level categorical indicator for

labor force status using a keyword search of the text fields "Occupation Current Text" and "Occupation Text_DC". This variable was coded as: industry employment, self-employment, disability, retirement, student, unemployment (including decedents who were incarcerated), and unknown/missing.

**Quantitative psychosocial circumstances.** The NVDRS includes a wide range of "circumstance variables" that encode correlates of suicidal behavior and precipitating events or behaviors flagged as noteworthy in source documents; each of these "circumstance" variables is coded as *Yes* vs. *No/Not Available/Unknown*.(16) To operationalize these correlates, we created dichotomous summary indicators of the following circumstance variables (each coded *Any* vs. *None or Not Available/Unknown*): 1) Socioeconomic problems (i.e., financial problem, job problem, or eviction/loss of housing), 2) Any physical health problem, 3) Bereavement (i.e., non-suicide death of friend/relative, suicide death of friend/relative), 4) Interpersonal problems (i.e., intimate partner, family relationship, other relationships, recent argument), 5) Alcohol/substance use (i.e., alcohol misuse, substance misuse, recent release from a rehabilitation facility), 6) Mental health problems (i.e., any mental health code, depressed mood, recent release from a psychiatric care facility), and, for those aged 10–24 years, 7) School problems. Additional details are provided in **S1 Appendix**.

**Qualitative narrative texts.** The CME and LE narratives are paragraphs of free text written by abstractors using information from original source documents (e.g., medical examiner records, law enforcement reports) which are designed to provide contextual details of the incident as well as salient circumstances of the decedent's life at the time of death (15). These texts contain information summarized across all data sources available to the abstractors, and they typically reflect the quantitative variables in the dataset. However, they are also where abstractors provide additional contextual information that may not be represented in the quantitative variables, which are a pre-defined set of circumstances. In the 2020 data used in this analysis, the average word count of CME and LE narratives were 152 words (SD: 107) and 119 words (SD: 117), respectively.

**Case definition of 'pandemic-related circumstances' from the narrative texts.** Since 2020, NVDRS narratives have incorporated information about pandemic-related circumstances (e.g., concerns about infection, job losses due to workplace closures) [16]. However, the manner and details around COVID-19 pandemic lacked consistency, especially in the beginning stages of the pandemic. For example, some states systematically included the details of the COVID-19 testing procedures (i.e., whether decedent was tested or what test results were), while other states only described COVID-19-related challenges that decedents experienced prior to their death. In some cases narratives simply mentioned COVID testing or a testing result, often recorded in the text as simply "COVID: positive" or "COVID: negative" without any additional context, In these cases there was to know whether the decedent was *aware* that they had COVID at the time of death, whether the positive test result was *salient* to their death, or even whether the testing occurred pre- or *post-mortem* (and therefore could not be reasonably considered as a 'contributing' circumstance).

To address potential inconsistencies around reporting pandemic-related circumstances, we employed an iterative, data-driven approach to defining PrC from the narrative texts. First, using keywords, we read examples of narratives that had the term "COVID" to form an understanding of how the pandemic was described in these texts. This immersion step was necessary to develop a case definition that was empirically-grounded in the data we intended to apply it to. Next, we developed a case definition that for a narrative to be coded as having PrC, the text had to *explicitly describe that some distress, event, or precipitating behavior in the decedent's life was connected to the onset or continuation of COVID-19 pandemic.*

Examples of narratives which we coded as having PrC using this definition included those which described decedents' fears/worries about getting COVID, restrictions to control viral

spread, or the government response to the pandemic; decedents' behaviors regarding COVID i.e., self-imposing extreme isolation measures; or distress and/or agitation regarding personal circumstances directly impacted by the pandemic (i.e., job loss due to COVID, financial strain due to COVID). These narratives would often either directly mention pandemic, indirectly refer to "the current circumstances" or "the current state of the world" (understood to include the pandemic given that their death occurred during the pandemic), or refer to circumstances that were unique to the early phase of the pandemic (e.g., transitions to remote schooling, social distancing).

Our PrC case definition excluded instances where the narrative included keywords related to the pandemic, but the text itself did not explicitly describe those keywords in connection to the death. For example, narratives were not coded as having PrC if the text simply mentioned the results of the decedent's COVID test without additional context indicating that this test result was a relevant circumstance (e.g., "tested positive for COVID" or "COVID: Negative" were not automatically coded as PrC). Narratives were also not coded as having PrC if the text described a circumstance that may have been connected to the pandemic but the text did not explicitly link this event to the pandemic (e.g., a job loss that took place during the early phase of the pandemic (i.e., April 2020), but without text explicitly describing that this job loss was connected to the pandemic, was not coded as being a PrC).

**Applying this definition to identify cases with pandemic-related circumstances from the narrative texts.**   We appended the CME and LE narratives (combined median character length: 1,286; Range: 47 to 19,549). We used a multi-step process to identify PrC from this joint text as described in our case definition. We first parsed the narratives using an initial set of 21 key words/phrases, identified by the research team, which signified the pandemic (e.g., "covid", "pandemic", "lockdowns", "quarantining"), which identified 2,876 narratives. We also manually reviewed the narratives of cases in which the NVDRS abstractors had indicated they were positive for '*Disaster exposure*' (this variable was modified by the NVDRS staff to include the pandemic but not until November 2020, per NVDRS staff). There were 94 cases that were 'PrC-negative' according to our initial set of keywords but were positive for the '*Disaster exposure*' variable, and six of these narratives described PrC according to our case definition. These six cases were added to the 'PrC-positive' group. Second, we augmented this set of words/phrases using text analysis tools (i.e., word2vec) [21] to identify additional terms that are semantically similar to ours (e.g., "e-learning", "distance learning", "shelter-in-place orders"), but not already in our list; this step which identified an additional 25 cases. Finally, we identified a set of exclusion phrases which conveyed information about the pandemic, but did not describe PrC (e.g., "covid: negative," "covid: no") for recoding as 'PrC-negative.' We note that we considered narratives that contained *both* pandemic-related terms *and* exclusion phrases (e.g., text reported both that "decedent struggled with distance learning" and "covid: negative") as being 'positive' for PrC.

Multiple independent annotators then manually reviewed 500 cases to assess inter-annotator agreement and evaluate the accuracy of this keyword/phrase approach to identifying PrC (detailed in next section). Using this approach, we identified n = 2,502 cases as having PrC (**Fig 2**). We coded the final PrC indicator variable as *Yes* vs. *No/Not Available/Unknown* to match the other circumstance variables in the NVDRS. Additional details regarding the case identification process are in the **S2 Appendix.**

## Validation of our approach to identifying pandemic-related circumstances

There is no existing "gold standard" label to validate our PrC variable label against in the NVDRS data. Therefore, we evaluated the performance of our case identification approach using extensive manual review by independent raters. To begin, three independent annotators

| 2020 Analytic Sample (n = 35,861) | → | Initial PrC Keyphrases (n = 2,876) | → | Augmented PrC Keyphrases (n = 2,901) | → | Exclusion Keyphrases (n = 2,502) |
|---|---|---|---|---|---|---|

**Fig 2. Overview of process for identifying pandemic-related circumstances in the narrative texts.** Among the 35,861 decedents included in our analytic sample, we applied our definition of pandemic-related circumstances (PrC) to identify cases from the narrative texts. We first parsed the narratives using an initial set of key words/phrases, identified by the research team, which signified the pandemic (e.g., "covid," "pandemic," "lockdowns"), which identified 2,876 narratives. This set was then augmented with words/phrases using text analysis tools to identify additional terms that were semantically similar (e.g., "distance learning," "shelter-in-place orders"), which identified an additional 25 narratives. Finally, we identified a set of exclusion phrases which conveyed information about the pandemic but in a manner that did not meet our case definition of PrC (e.g., "covid: negative," "covid: no"). Trained annotators then independently manually reviewed 500 cases (250 PrC 'positive' and 250 PrC 'negative') to assess inter-annotator agreement and evaluate the accuracy of this approach to identifying PrC.

labeled subsets of 'PrC-positive' and 'PrC-negative' cases, as identified from the key word/phrase approach described above. Initially, inter-annotator agreement was calculated on a set of 100 narratives (a random sample of 50 'PrC-positive' and a random sample of 50 'PrC-negative' cases). Discordant labels were discussed and resolved. Inter-annotator agreement was high (Krippendorff's alpha = 0.90), and therefore the remaining cases were each independently labeled by two annotators. Again, any discordant labels were discussed and resolved before calculating positive and negative predictive value of the case identification approach.

In the random sample of 250 'PrC-positive' narratives, there were 19 cases that the annotators identified as 'false positives'. Assuming manual review of the narrative as the gold-standard, the positive predictive value (PPV) of our PrC case identification approach was calculated as 0.92. Of the 250 'PrC-negative' cases reviewed by the annotators, none were determined to have pandemic-related circumstances (i.e., zero 'false negatives'). Again, assuming manual review of the narrative as the gold-standard, this generated a negative predictive value (NPV) of 1.0.

However, since we expect that the 'true' prevalence of PrC cases to be low, the likelihood of zero false negatives in a random sample of 250 cases (out of >30,000 total narratives) is not unlikely to occur by chance (i.e., the true NPV may be less than 100%). Therefore, we conducted further validation using manual review by evaluating two subsets of 'PrC-negative' cases that we felt were more likely to contain false negatives. The first subset was a random sample of 100 'PrC-negative' cases that contained phrases similar to our PrC inclusion phrases, but which lacked the explicit description of the pandemic required by our case definition (e.g., a phrase like 'a bout with a virus'). The second subset was a random sample of 100 'PrC-negative' cases that had characteristics that are conceptually independent of, but empirically correlated with, being 'PrC-positive'; these characteristics included a) date of death between March and May, b) narrative length >1,286 characters, and c) NVDRS annotators coded the case as having 'financial problems' in their contributing circumstances. Manual review of these 200 cases by two independent annotators identified 2 that had pandemic-related circumstances (i.e., two false negatives), for an estimated NPV of 0.99. Finally, we augmented this manual review by using a Bayesian model to calculate the expected distribution of the NPV of the PrC identification procedure, using the original random sample of n = 250 'PrC-negative' cases (all of which were 'true negatives' after manual review, as described above) to model a simulation (details provided in the **S2 Appendix**). Using this Bayesian model, we calculated the NPV = 0.99 [95% CI: 0.98, 0.99].

## Analysis

We used $X^2$ tests for categorical variables and t-tests for continuous variables to compare the characteristics of decedents with PrC to those without in 2020, and to compare the characteristics of all suicide deaths in 2020 to those in the pre-pandemic period. **S1 Table** provides additional descriptive variables of the analytic samples.

**Time series analysis of suicide deaths over 2020.** Given the dynamic nature of the pandemic, we initially sought to visualize the temporal trends of suicide deaths in 2020 compared to the pre-pandemic period. We conducted a time series analysis by directly standardizing the 4-week rolling averages in 2020 to the corresponding average case counts of 2017/2018. Values <1 indicate that 2020 had fewer deaths than the pre-pandemic period in that week, and values >1 indicate that 2020 had more deaths. This procedure accounts for seasonality in the baseline period suicide risk and the effects of other temporally-varying factors by showing 2020 mortality relative to mortality in the pre-pandemic period. This procedure accounts for temporal autocorrelation by comparing 4-week moving averages rather than weekly suicide frequencies. Additional details regarding how we created the visualizations of these trends is provided in **S3 Appendix**. Since narratives with PrC tended to be longer (**S1 Table**), we performed 'full matching' [22] of the 2020 and pre-pandemic samples by the narrative length, numbers of quantitative variables marked as "unknown," and whether an autopsy was performed, to account for potential data quality differences.

**Relationship between PrC with other psychosocial circumstances.** Using the 2020 data only, we examined the relationship between other psychosocial circumstances (the seven circumstance groupings described above, plus their components) with PrC using logistic regression. The outcome of these models was 'PrC' vs. 'no-PrC.' Models were adjusted for calendar quarter of the year, state, sex, age group, race, marital status, education, and employment status. Confidence intervals and p-values were adjusted for multiple hypotheses using a Holm-Bonferroni correction.

**Topic modeling of narrative texts that had pandemic-related circumstances.** To explore and summarize the ways in which PrC were described in the narratives, we applied a contextualized topic model [23] to these texts. Topic models use both individual words in each sentence and a high-dimensional vector representation of the meaning of each text 'token' to identify themes [23,24]. Complete details regarding data preparation, tokenization, labeling, refinement, and software packages used for the topic modeling are provided in **S4 Appendix**. Briefly, we began with an initial labeling of sentences which was followed by a combination of unsupervised and supervised topic models [24]. The model generates estimates of each topic probability i.e., weighted fraction of the narrative's words that are associated with each topic, in which words are down-weighted if they are common across multiple topics. We used a topic probability threshold of >0.10 to parse topics for extraction (85% of potential topics met this threshold), which provided a sufficiently heterogeneous set of topics to examine. Across multiple iterations that examined interpretability, consistency, and probability of extracted topics, we identified 11 topics from the narratives that summarized the subject content and emotional salience of these texts. **S2 Table** provides the frequencies, strongest associated words, and exemplary excerpts from the narrative texts for each of these topics.

Finally, to explore demographic variation in topic frequency, we regressed the topic probability on age, race, sex, marital status, and education level using a beta regression. Coefficients from this regression represent differences in the log-odds of each topic associated with the demographic characteristic; 95% confidence intervals for each coefficient were adjusted for multiple comparisons using the Holm-Bonferroni correction.

Analyses were conducted using R (v3.6.3) and Python (v3.9.12), and all p-values refer to two-tailed tests.

# Results

## Characteristics of decedents that had pandemic-related circumstances

As shown in **Table 1**, 2,502 (6.98%) of the narratives of suicide deaths in 2020 had PrC. Decedents with PrC were significantly older (e.g., 23% vs. 17.7% were aged ≥65), were more likely

**Table 1. Decedent characteristics by pandemic vs. pre-pandemic periods, and by presence of pandemic-related circumstances (PrC) in the case narratives in 2020.**

| | Pre-Pandemic vs Pandemic Time Periods | | | Year 2020 Only | | |
| --- | --- | --- | --- | --- | --- | --- |
| | Years: 2017–2018 | Year: 2020 | p-value | Narratives that did not have PrC | Narratives that had PrC | p-value |
| **Total Sample** | N = 61,019 | N = 28,001 | | N = 33,359 | N = 2,502 | |
| Single Suicide | 53625 (87.9%) | 24303 (86.8%) | | 29439 (88.2%) | 2328 (93.0%) | |
| Undetermined Intent | 7394 (12.1%) | 3698 (13.2%) | | 3920 (11.8%) | 174 (7.0%) | |
| **Age Groups** | | | <0.001 | | | <0.001 |
| 10–24 | 8258 (13.5%) | 3837 (13.7%) | | 4578 (13.7%) | 341 (13.6%) | |
| 25–44 | 20592 (33.7%) | 9888 (35.3%) | | 12076 (36.2%) | 695 (27.8%) | |
| 45–64 | 22081 (36.2%) | 9230 (33.0%) | | 10807 (32.4%) | 890 (35.6%) | |
| 65+ | 10088 (16.5%) | 5046 (18.0%) | | 5898 (17.7%) | 576 (23.0%) | |
| **Sex** | | | <0.001 | | | 0.006 |
| Male | 46216 (75.7%) | 21737 (77.6%) | | 26006 (78.0%) | 1891 (75.6%) | |
| Female | 14801 (24.3%) | 6263 (22.4%) | | 7352 (22.0%) | 611 (24.4%) | |
| **Race and Ethnicity** | | | <0.001 | | | <0.001 |
| White | 48879 (80.1%) | 21407 (76.5%) | | 25700 (77.1%) | 1973 (79.0%) | |
| American Indian/Alaska Native | 689 (1.1%) | 386 (1.4%) | | 467 (1.4%) | 22 (0.9%) | |
| Asian/Pacific Islander | 1496 (2.5%) | 734 (2.6%) | | 834 (2.5%) | 114 (4.6%) | |
| Black/African American | 4582 (7.5%) | 2570 (9.2%) | | 2950 (8.9%) | 139 (5.6%) | |
| Hispanic/Latino | 4624 (7.6%) | 2466 (8.8%) | | 2896 (8.7%) | 211 (8.5%) | |
| Multiracial/Other | 745 (1.2%) | 402 (1.4%) | | 471 (1.4%) | 38 (1.5%) | |
| **Education Status** | | | 0.243 | | | <0.001 |
| 8th grade or lower | 2121 (3.6%) | 1007 (3.7%) | | 1180 (3.6%) | 113 (4.6%) | |
| 9-12th grade | 7617 (12.8%) | 3467 (12.6%) | | 4142 (12.7%) | 243 (9.9%) | |
| High School or GED | 24519 (41.3%) | 11516 (42.0%) | | 13970 (42.7%) | 839 (34.1%) | |
| Some college | 9993 (16.8%) | 4593 (16.8%) | | 5500 (16.8%) | 409 (16.6%) | |
| Associate's | 4449 (7.5%) | 1955 (7.1%) | | 2341 (7.2%) | 200 (8.1%) | |
| Bachelor's | 7211 (12.1%) | 3245 (11.8%) | | 3761 (11.5%) | 410 (16.7%) | |
| Master's | 2388 (4.0%) | 1138 (4.2%) | | 1242 (3.8%) | 166 (6.8%) | |
| Doctorate | 1082 (1.8%) | 488 (1.8%) | | 555 (1.7%) | 77 (3.1%) | |
| Unknown or Missing | 1638 (2.7%) | 592 (2.1%) | | 668 (2.0%) | 45 (1.8%) | |
| **Relationship Status** | | | <0.001 | | | <0.001 |
| Married/Partnered | 17453 (29.0%) | 7645 (27.7%) | | 9255 (28.1%) | 827 (33.3%) | |
| Separated/Divorced/Widowed | 17898 (29.7%) | 7910 (28.6%) | | 9493 (28.8%) | 681 (27.4%) | |
| Single/Never Married | 24890 (41.3%) | 12075 (43.7%) | | 14186 (43.1%) | 979 (39.4%) | |
| **Employment Status** | | | <0.001 | | | <0.001 |
| Industry Employment | 11568 (19.0%) | 5146 (18.4%) | | 5779 (17.3%) | 539 (21.5%) | |
| Self-employment | 2500 (4.1%) | 1245 (4.4%) | | 1495 (4.5%) | 121 (4.8%) | |
| Disability | 1676 (2.7%) | 589 (2.1%) | | 702 (2.1%) | 42 (1.7%) | |
| Retirement | 2001 (3.3%) | 799 (2.9%) | | 817 (2.4%) | 114 (4.6%) | |
| Student | 3905 (6.4%) | 1533 (5.5%) | | 1737 (5.2%) | 194 (7.8%) | |
| Unemployment | 5195 (8.5%) | 2198 (7.8%) | | 2253 (6.8%) | 293 (11.7%) | |
| Unknown or Missing | 39624 (64.9%) | 19074 (68.1%) | | 23677 (71.0%) | 1404 (56.1%) | |

**Note**: The 2020 sample was restricted to cases with known circumstances and joint LE/CME narratives that had a length of ≥35 characters.

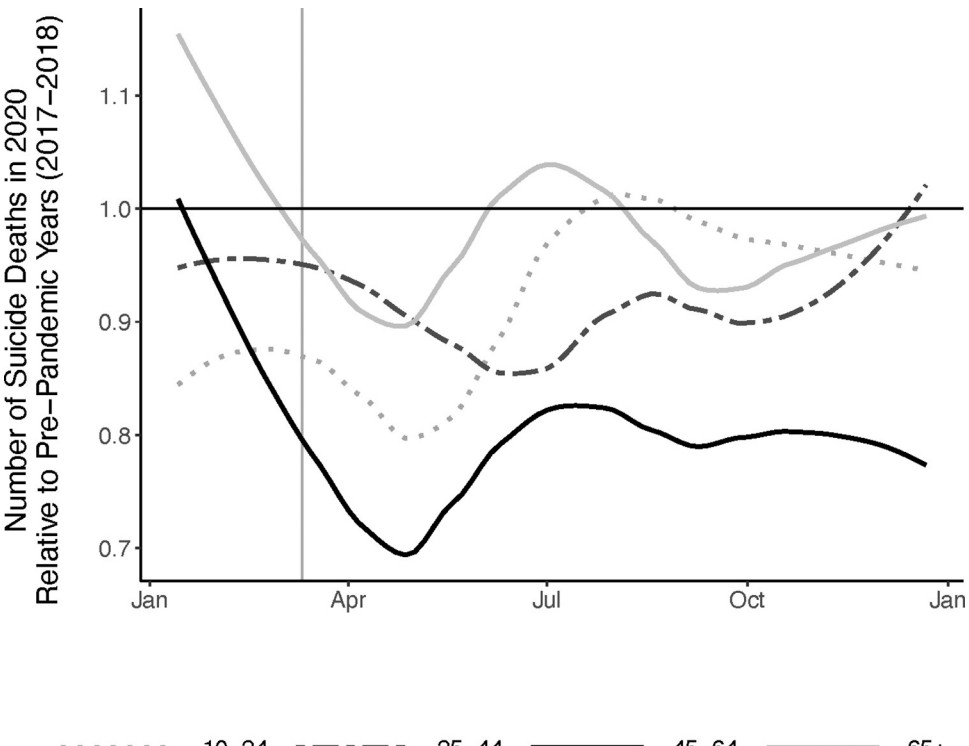

**Fig 3. Temporal trend in suicides deaths in 2020 relative to the pre-pandemic period by age group.** The x-axis of is months of the year. The y-axis is the ratio of suicide deaths during each 4-week period in 2020 (n = 25,612) relative to the same 4-weeks of the pre-pandemic period 2017–2018 (n = 57,484) [total n = 83,090].

to be female (24.4% vs. 22.0%), non-Hispanic white (79.0% vs. 77.1%), married (33.3% vs. 28.1%), and highly educated (e.g., 9.9% vs. 5.5% had an advanced degree) compared to decedents in 2020 that did not have PrC. Decedents with PrC were also more likely to be unemployed compared to other decedents in 2020 (11.7% vs. 6.8%); however, substantial missingness on employment status warrants caution.

**Variation in suicide deaths over the 52 weeks of 2020.** S1 Fig illustrates the number of suicide deaths per week in 2020 and in the pre-pandemic period (2017–2018). It shows, as has been previously reported [11,25], that the number of suicide deaths in 2020 was nearly identical to the pre-pandemic period until March, when the count of suicide deaths declined sharply. This replication is reassuring given that the NVDRS data analyzed here is limited to 37 states, to suicide decedents with known circumstances, and that our comparison period excludes 2019 due to data quality concerns, as discussed above. Fig 3 displays the weekly count of suicide deaths in 2020 standardized to the pre-pandemic period by age group. It illustrates that the 45–64 age group drove the spring 2020 decline in suicide counts, with more modest deviations from the pre-pandemic period among other ages. We explored alternative specifications of the pre-pandemic period (S2 Fig), and our results were robust to these alternative comparisons. S3 Fig shows that in 2020, deaths with PrC in the narrative were more likely to occur in April/May, with lower likelihood of occurring later in the year, relative to deaths that did not have PrC.

**Relationship between pandemic-related and other psychosocial circumstances.** Next, we examined the relationship psychosocial circumstances, as indicated by the seven groups of NVDRS variables, with PrC. As shown by Table 2, decedents with PrC were more likely to

**Table 2. Psychosocial circumstances related to pandemic-related circumstances (PrC): NVDRS 2020.**

| | PrC in Narrative (n = 2,161) | | No PrC in Narrative (n = 28,781) | | Adjusted Odds Ratio [95% CI] |
|---|---|---|---|---|---|
| | N | Proportion | N | Proportion | PrC vs. no-PrC |
| **Any socioeconomic problem** | 755 | 0.349 | 3,619 | 0.126 | 3.577* [3.032, 4.216] |
| Financial problem | 338 | 0.156 | 1,649 | 0.057 | 2.696* [2.175, 3.324] |
| Job problem | 609 | 0.282 | 2,057 | 0.071 | 5.08* [4.2, 6.135] |
| Eviction/Loss of home | 76 | 0.035 | 759 | 0.026 | 1.282 [0.901, 1.782] |
| **Any physical health problem** | 622 | 0.288 | 6,115 | 0.212 | 1.237* [1.053, 1.450] |
| **Any bereavement** | 243 | 0.112 | 2,042 | 0.071 | 1.458* [1.158, 1.82] |
| Non-suicide death of family/friends | 194 | 0.090 | 1,581 | 0.055 | 1.570* [1.211, 2.014] |
| Suicide death of family/friends | 60 | 0.028 | 521 | 0.018 | 1.186 [0.814, 1.679] |
| **Any interpersonal problem** | 590 | 0.273 | 9,012 | 0.313 | 0.791* [0.677, 0.922] |
| Intimate Partner Problem | 402 | 0.186 | 6,825 | 0.237 | 0.719* [0.597, 0.863] |
| Family Relationship | 155 | 0.072 | 1,795 | 0.062 | 1.003 [0.836, 1.195] |
| Other Relationship Problems | 60 | 0.028 | 540 | 0.019 | 1.386 [0.915, 2.033] |
| Argument | 283 | 0.131 | 4,332 | 0.151 | 0.856 [0.706, 1.032] |
| **Any alcohol/substance problem** | 664 | 0.307 | 10,966 | 0.381 | 0.795* [0.683, 0.925] |
| Alcohol misuse | 469 | 0.217 | 5,949 | 0.207 | 1.007 [0.884, 1.144] |
| Substance misuse | 312 | 0.144 | 7,275 | 0.253 | 0.594* [0.481, 0.728] |
| Recent release from rehabilitation facility | 8 | 0.004 | 138 | 0.005 | 0.931 [0.335, 2.083] |
| **Any mental Health problem** | 1,637 | 0.758 | 17,253 | 0.599 | 1.788* [1.524, 2.106] |
| Any mental health diagnosis | 1,201 | 0.556 | 13,292 | 0.462 | 1.241* [1.082, 1.425] |
| Depressed mood | 1,061 | 0.491 | 8,106 | 0.282 | 2.337* [2.021, 2.703] |
| Recent release from psychiatric care facility | 56 | 0.026 | 519 | 0.018 | 1.289 [0.864, 1.865] |
| **Any school problem** | 74 | 0.217 | 299 | 0.065 | 3.14* [1.857, 5.233] |

Psychosocial circumstances reflect from NVDRS quantitative variables. PrC reflects narrative texts, as described in the S2 Appendix. School problems limited to decedents aged 10–24 (n = 4,919).

Odds ratios adjusted for state, quarter of the year, autopsy status, age, sex, race, marital status, education, and employment status. *Indicates ORs with Holm-Bonferroni adjusted p-value<0.05.

have circumstances related to financial and job problems, physical health problems, non-suicide related bereavement, mental health problems, and, for young adults, school problems relative to decedents who did not have PrC. However, decedents with PrC were less likely to have circumstances related to alcohol/substance use problems or intimate partner problems relative to those who did not have PrC. Finally, circumstances related to housing loss, suicide death of family/friends, other interpersonal relationships, and recent discharge from psychiatric or rehabilitative facilities were not significantly correlated with PrC.

## Exploring topics represented in the PrC narratives

The 11 topics identified in the narratives with PrC fell into four broad, non-mutually-exclusive groupings: (1) Public health containment measures and related concerns [topics in this grouping: *COVID-19 testing and isolation* (prevalence: 19.9%), *COVID-19 restrictions* (prevalence: 19.3%), and *Quarantine, movement, and change of space* (prevalence: 16.9%)]; (2) Changing social and economic situations [topics in this grouping: *Job or business loss* (prevalence: 14.3%), *Unstable work, financial or family environment* (prevalence: 18.1%), and *Remote schooling and social adjustment* (prevalence: 11.2%)]; (3) Physical health problems and concerns [topics in this grouping: *COVID-19 testing and isolation* (prevalence: 19.9%) and *COVID symptoms and other health problems* (prevalence: 17.9%)]; and (4) Psychological distress and concerns [topics in this grouping: *Stress and problems* (prevalence: 11.5%), *Fear and frustration* (prevalence: 20.6%), *Mental health symptoms exacerbated by the pandemic* (prevalence: 12.7%), and *Isolation and related anxiety* (prevalence: 18.0%)]. See **S2 Table** for exemplar narrative excerpts.

**Fig 4** arrays these 11 topics by age group. Narratives of decedents aged 10–24 that had PrC most often described challenges related to *schooling*, including adjustment to remote learning and *social isolation*, including separation from peers and being unable to participate in activities. Narratives of decedents aged 25–64 most often described challenges related to *unstable work/financial situations* and *job or business loss* (e.g., lost employment, closed businesses, fears around financial instability, remote work). Narratives of decedents aged 65+ were most likely to describe concerns around risk of infection and experiencing effects of the pandemic (i.e., *COVID testing, COVID symptoms)*; they also frequently referenced *fear and frustration or isolation and anxiety* (i.e., uncertainty about current events). The panels of **S4 Fig** illustrate the relationship between a core set of demographic characteristics, including age group, sex, race, education and marital status, and the log-odds of each topic. In these forest plots, the coefficient estimates are shown in blue if they are significant and positive, in red if they are significant and negative, and in gray if they are not statistically significant. This plot again emphasizes that there is notable variation in the prevalence of several topics as a function of age groups, with fewer differences across other demographic characteristics.

## Discussion

In 2020, approximately 7% of suicide deaths reported in the NVDRS did reference some aspect of the pandemic as a contributing circumstance. Deaths with PrC primarily occurred during the spring and early summer months (47% occurred between March and June). Suicide decedents with PrC tended to be older and more educated relative to those whose narratives did not include PrC. PrC were positively associated with circumstances related to work and financial problems, school problems, mental health problems, and physical health problems; however, PrC were inversely associated with circumstances of interpersonal problems and substance misuse. Topic modeling of the PrC narratives indicates that these texts most frequently discussed the pandemic in terms of concerns about the economy/work, social

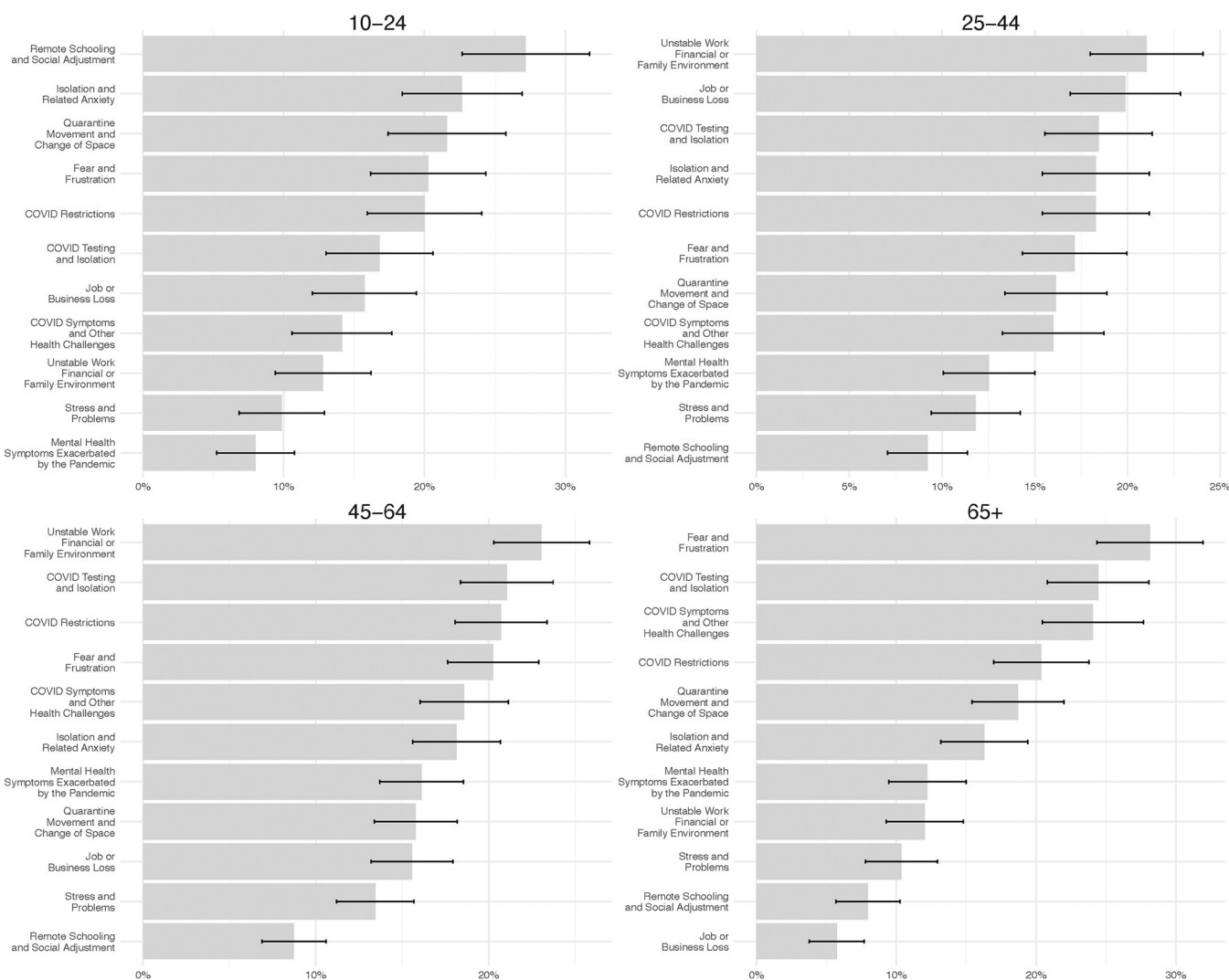

**Fig 4. Most common topics described in the narratives that had pandemic-related circumstances: 2020 NVDRS.** The x-axis represents the proportion of cases with pandemic-related circumstances (PrC) that are mentioned in each specific topic. Limited to cases in 2020 that had CME/LE narratives ≥35 characters long (n = n = 35,861). See S4 Appendix and S2 Table for additional details.

distancing restrictions, risk of infection, and other social disruptions. Taken together, these findings provide the most comprehensive description to date of how pandemic-related circumstances salient to suicide mortality varied during the first year of the pandemic.

We confirmed that the US experienced a modest decline in the overall rate of suicide in 2020 from pre-pandemic levels, as reported previously [11]. This analysis provides new information showing that adults aged 45–64 drove this decline in March/April, which persisted through 2020; other age groups also experienced the spring decline, but rates in these groups returned to pre-pandemic rates by late summer (<25 years and 65+ years) or the end of the calendar year (25–44 years). These findings most closely match those of Min et al. (2022), who found that the decline in suicide during 2020 in Korea was driven largely by adults aged 40–64 [26]. While beyond the scope of this analysis, it is important to contextualize these findings within the international literature examining how rates of suicide varied over the first 12 months of the pandemic; with few exceptions, in most nations where this question has been

explored, the overall rate of suicide either modestly declined or was unchanged during this acute phase of the pandemic relative to prior years [27], although not all studies have found that this decline varied by age.

Although emerging studies have begun examining changes in suicide rates in relation to specific pandemic-control policies (e.g., school reopening and suicide among adolescents) [28], our analysis shows that a robust description of how suicide risk varied over time is required before conclusions can be drawn from such efforts. Researchers do not yet have a understanding of the factors that drive preexisting variation in suicidal behavior (i.e., by age, place, sex, race, etc.) [29]. This knowledge gap impairs our ability to specify testable alternative hypotheses, namely: *Should we have anticipated an absolute decrease in the suicide rate during the pandemic, and if so, for whom and for how long*? It also poses challenging scientific questions, such as: *Given the wealth of data indicating substantial increases in emotional distress and suicidal ideation during the first year of the pandemic, what does the fact that suicide mortality not increase mean about the processes underlying emotional distress, suicidal ideation, and suicide mortality*? Clarity regarding these expectations is needed before researchers can meaningfully interpret the impacts of specific policy changes during the pandemic, both during 2020 and beyond.

While the acute phase of the pandemic did not result in an overall increase in suicide mortality in the US, this does not mean that the pandemic was irrelevant to the suicide deaths that occurred in 2020. While narratives described PrC, decedents were 3.5 times more likely to also have socioeconomic-related circumstances, particularly problems with jobs. Job strain and loss are established risk factors for suicide [30], and stay-at-home-orders and "essential worker" designations [31] dramatically impacted work in the early months of the pandemic. While the CARES Act (passed by Congress in March 27, 2020) provided individual stipends and grants for businesses, additional funding for unemployment insurance and paid medical leave, and launched programs to help employers prevent layoffs, these policies took time to actually implement, varied in eligibility criteria, and not all states adopted all programs [32]. Many of these work and financial assistance programs were under-utilized [33], and thus it is not surprising that the financial benefit of these programs was not experienced equally by all Americans [34]. Situating the findings from this study within the broader literature on how policies related to employment shifted during the pandemic, our findings suggest that policy-makers and public health practitioners partner with employers, community organizations, unions, and trade groups to ensure timely outreach, clear communication, and empathetic messaging about programs to address the financial consequences of pandemic control measures [35].

Decedents with PrC were also more likely to have physical health and mental health problems relative to those that did not have PrC. While the NVDRS circumstance variables do not distinguish between long-standing and acute health challenges, the topic modeling showed that themes related to testing and/or isolating due to COVID-19 exposure and experiencing symptoms related to COVID-19 were represented in approximately 20% of the PrC narratives. This illustrates that individuals infected with COVID-19 may benefit from diverse types of support (e.g., emotional, social, as well as medical) from their healthcare providers. Given the urgent need for physicians and nurses in hospitals during pandemics, this emphasizes a role for different types of providers that operate *outside* the clinic, such as home health aides and community health workers (CHW) [36], to address these needs; indeed, several states did expand the role of CHWs as part of their pandemic response programs [37]. Recent studies have shown that programs that use these types of workers to engage with people hospitalized for COVID improves depressive symptoms and overall quality of life [38]. As the lingering and diverse effects of COVID infection are better understood [39], the need for such integrative and comprehensive post-infection support and rehabilitation programs will likely grow.

## Limitations and strengths

First, we recognize that macro events are multifaceted (e.g., COVID-19 had social, economic, political, and health implications). Other temporally-salient factors (e.g., policies that targeted pandemic-related unemployment or housing loss) may have contributed to variation in suicide trends over the calendar year. However, we chose not to account for these types of contemporaneous factors in this analysis since the onset of the pandemic was associated with many sudden changes in these factors, making it challenging to account for their effects. By providing a comprehensive characterization of suicide mortality over the first year of the pandemic, these findings provide an empirical foundation from which future work can explore the impact of social, economic, and political factors on suicide mortality, particularly as the pandemic has moved beyond this emergency phase.

Second, most prior studies of suicide during the pandemic "assigned" exposure using ecological indicators (i.e., shutdown orders, COVID-19 case counts), which makes comparisons difficult, as our measures of PrC had greater specificity than these studies. Third, macro events may impact data quality; we chose not to include data for 2019 because it was abstracted during 2020, when states faced various degrees of shutdowns that impacted relationships with local stakeholders.(25) Fourth, important competing risks (e.g., accidental overdose, COVID-19 mortality) that affected both the population at risk and potentially case status (e.g., misclassification of suicide deaths as accidental overdose) also varied dynamically over 2020.

Finally, although narratives provide contextual details relevant to suicide deaths, they do not reveal "why" a suicide occurred. Rather, the intent of these texts is to describe and provide additional details regarding the circumstances that were, according to the data sources available to the NVDRS abstractors, present in the person's life and relevant to their death. This analysis relies on the assumption that pandemic-related circumstances, as defined in this study, were consistently and accurately recorded in the case narratives. However, due to biases in the data generation process, including in the processes of legal investigations and writing of the narratives by abstractors, information regarding pandemic-related factors may be incomplete from the narratives of some groups [17,40]. For instance, narratives that referenced PrC were, on average, ~300 characters longer than narratives that did not, implying that there was more *information potential* in the 'PrC-positive' texts. Others have shown that, irrespective of the pandemic (and using pre-2020 data), the NVDRS texts tend to be shorter among decedents who are Black or Asian/Pacific Islander, older, had less education, and are unmarried [41]. To account for these differences in information potential and the potentially undercounting of PrC in certain groups, we limited our analytic samples to decedents that had 'known' circumstances, complete data on demographic covariates, and had a minimal narrative length. However, even with these restrictions there may be under-reporting of PrC in certain groups, which could have influenced our topic modeling results. Indeed, in the 2020 data, ~20% of suicide decedents did not have 'known' circumstances. The CDC has argued that strong data-sharing partnerships are a critical component of NVDRS data quality, but also acknowledged that these partnerships have been impacted by the pandemic itself [25]. The net result is that we feel it is most appropriate to view the results of the topic modeling analysis as primarily hypothesis *generating* rather than hypothesis *testing*.

This study also has multiple strengths. The NVDRS is the most comprehensive registry of suicide deaths in the US. Unlike prior work that had inferred the role of COVID-19 or pandemic restrictions ecologically (e.g., timing of policies), our analysis directly identified cases in which PrC were salient using the narratives. The time-series analysis compared weekly rates of suicide in 2020 to the pre-pandemic period in a manner that accounts for temporal variation both in suicide and the emerging pandemic. Finally, by restricting these trend analyses to only

those states with an established history of reporting to the NVDRS, matching by information availability, and setting a threshold minimum length on the narratives for our analysis, we enhance the rigor of our comparisons both across time and across groups in 2020.

## Public health implications

The reverberating consequences of COVID-19 on the lives of individuals and communities is only beginning to be explored using rigorous research designs [33], and the full set of consequences for population mental health will likely not be known for many years [42]. The pandemic increased public awareness regarding mental health, creating an opportunity to explore new approaches to promoting emotional well-being and strengthen existing public mental health initiatives [43]. Our findings also emphasize the need for greater interprofessional and cross-sector collaboration and coordination to implement programs that support mental health during times of crisis, particularly with employers and related organizations. While the acute phase of the pandemic involved time-sensitive policies (that have now largely been sunset) to mitigate this emergency, it must be appreciated that these policies were not designed to address long-standing factors that contribute to suicide risk, such as social isolation, financial insecurity, emotional distress, pain and functional impairment [43]. Findings from this study show that these established correlates of suicide mortality form the context in which acute crises, like COVID-19, are experienced by individuals. Finally, despite calls by policy leaders to address this issue [44], the pandemic has worsened workforce shortages that significantly limit most Americans' ability to access timely and high-quality behavioral health care [43]. Addressing these psychosocial and infrastructure issues is critical to ensuring that communities and the public mental health workforce are better prepared to respond to future crises and emergencies.

## Supporting information

**S1 Fig. Number of suicides by week in 2020 vs. the pre-pandemic period.**
(DOCX)

**S2 Fig. Sensitivity analysis examining robustness of the time-series analysis to alternative pre-pandemic reference periods.**
(DOCX)

**S3 Fig. Relative number of deaths in 2020 by whether the case narrative described pandemic-related circumstances.**
(DOCX)

**S4 Fig. Relationship between demographic characteristics and topic prevalence.**
(DOCX)

**S1 Table. Additional sample characteristics during the pre-pandemic (2017–2018) and pandemic (2020) time periods.**
(DOCX)

**S2 Table. Topic modeling of narratives with pandemic-related circumstances, including exemplar words and phrases.**
(DOCX)

**S1 Appendix. Methods: Details regarding NVDRS quantitative variables used in this analysis.**
(DOCX)

**S2 Appendix. Methods: Details regarding identification of Pandemic-related Circumstances (PrC) from the text narratives.**
(DOCX)

**S3 Appendix. Methods: Time-series analysis and visualization of suicide deaths over the 12 months of 2020.**
(DOCX)

**S4 Appendix. Methods: Topic modeling of the narrative texts that had pandemic-related circumstances.**
(DOCX)

## Author Contributions

**Conceptualization:** Briana Mezuk, Viktoryia Kalesnikava, Aparna Ananthasubramaniam, Sarah Burgard.

**Data curation:** Aparna Ananthasubramaniam.

**Formal analysis:** Viktoryia Kalesnikava, Aparna Ananthasubramaniam, Annalise Lane, Alejandro Rodriguez-Putnam, Lily Johns.

**Funding acquisition:** Briana Mezuk, Kara Zivin.

**Methodology:** Briana Mezuk, Viktoryia Kalesnikava, Aparna Ananthasubramaniam.

**Project administration:** Briana Mezuk.

**Visualization:** Aparna Ananthasubramaniam.

**Writing – original draft:** Briana Mezuk, Viktoryia Kalesnikava, Aparna Ananthasubramaniam.

**Writing – review & editing:** Briana Mezuk, Courtney Bagge, Sarah Burgard, Kara Zivin.

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
