## [Editor Report · Decision Letter 0]

4 Jun 2024

PONE-D-24-21265Psychosocial and Pandemic-Related Circumstances of Suicide Deaths in 2020: Evidence from the National Violent Death Reporting SystemPLOS ONE

Dear Dr. Mezuk,

Thank you for submitting your manuscript to PLOS ONE. After careful consideration, we feel that it has merit but does not fully meet PLOS ONE’s publication criteria as it currently stands. Therefore, we invite you to submit a revised version of the manuscript that addresses the points raised during the review process.

I commend you on the well-researched paper on an important topic.  However, to expedite the review process and ensure the paper meets the highest standards, I suggest revising it according to the comments below before resubmission.  This proactive step will likely save time on a round of reviews, and will facilitate a faster final acceptance.

We look forward to receiving your revised manuscript.

Kind regards,

Shrisha Rao, Ph.D.

Academic Editor

PLOS ONE

Journal Requirements:

"Funding: This project was supported by grants from the National Institute of Mental Health (R01-MH128198, to MPIs Mezuk and Zivin) and the American Foundation for Suicide Prevention (DIG-1-110-19, to Mezuk). The funders had no role in the conceptualization, analysis, interpretation, or decision to publish this manuscript."

"This project was supported by grants from the National Institute of Mental Health (R01-MH128198, to MPIs Mezuk and Zivin) and the American Foundation for Suicide Prevention (DIG-1-110-19, to Mezuk). The funders had no role in the conceptualization, analysis, interpretation, or decision to publish this manuscript."

4. Please note that your Data Availability Statement is currently missing the repository name and the DOI/accession number of each dataset or a direct link to access each database. If your manuscript is accepted for publication, you will be asked to provide these details on a very short timeline. We therefore suggest that you provide this information now, though we will not hold up the peer review process if you are unable.

**Additional Editor Comments:**

This paper analyzes suicide deaths in the U.S. in 2020 using data from the National Violent Death Reporting System (NVDRS).  It identifies cases where pandemic-related circumstances (PrC) were mentioned in the case narratives, and compares the characteristics of suicide decedents with PrC to those without. The authors also use topic modeling to identify common themes in the PrC narratives. Key findings presented include:

(i) 6.98% of suicide deaths in 2020 mentioned PrC.

(ii) Decedents with PrC tended to be older, more educated, and more likely to have financial problems and mental health issues compared to those without PrC.

(iii) Common PrC themes included concerns about pandemic restrictions, financial losses, infection risk, and social disruption.

(iv) Adults aged 45-64 drove an overall decline in suicide rates in spring 2020 compared to pre-pandemic years.

The following points can be addressed to improve the work.

1. The analysis relies heavily on the identification of PrC from case narratives, but the validation of this approach is limited.  More rigorous evaluation of the PrC labeling method, such as having multiple raters independently code a larger subset of narratives, would strengthen confidence in the findings.  (Also see point #5 below.)

2. While the topic modeling provides interesting insights into PrC themes, the analysis is largely descriptive. Applying additional quantitative measures, such as calculating topic prevalence by demographic group, could enable statistical comparisons and uncover meaningful patterns.

3. The time series analysis comparing 2020 to pre-pandemic years is useful but appears somewhat simplistic. Employing more sophisticated time series modeling techniques that account for seasonality, autocorrelation, and other temporal factors could provide a more nuanced understanding of pandemic-related changes in suicide rates.

4. The conclusions drawn from the findings seem rather limited.  The authors could delve deeper into the implications of their results for suicide prevention efforts during future crises.  For example, what interventions might mitigate the elevated risks associated with financial strain and isolation?  How can messaging be tailored to different age groups given the varying PrC themes?

5. The study relies on the assumption that pandemic-related circumstances were consistently and accurately recorded in the case narratives.  If such PrC were sometimes omitted or incorrectly mentioned, this could lead to underestimation or overestimation of the prevalence and impact of PrC.  A particular case of interest would be if the narratives happen to be biased based on some protected attribute (age, race, gender, etc.), so that victims of a particular class are more/less likely to be described as subject to PrC.  The authors should thus discuss the potential limitations of using narrative data and how this might affect the interpretation of their findings.

6. Finally, the generalizability of the findings to suicides beyond 2020 is unclear. Acknowledging the unique circumstances of the pandemic's first year and the need for ongoing research as the crisis evolves would be valuable additions to the discussion.

In summary, while this study offers novel insights into the impact of the COVID-19 pandemic on suicide deaths, the analysis would benefit from more rigorous validation of key measures, expanded quantitative assessments, and further reflection on the implications of the findings for research and practice. Addressing these limitations will enhance the scientific and practical value of this work.

---

## [Author Response · Author response to Decision Letter 0]

17 Jul 2024

Please see the attached Response to Reviewers document.

---

## [Decision Letter · Decision Letter 1]

13 Sep 2024

PONE-D-24-21265R1Psychosocial and Pandemic-Related Circumstances of Suicide Deaths in 2020: Evidence from the National Violent Death Reporting SystemPLOS ONE

Dear Dr. Mezuk,

Thank you for submitting your manuscript to PLOS ONE. After careful consideration, we feel that it has merit but does not fully meet PLOS ONE’s publication criteria as it currently stands. Therefore, we invite you to submit a revised version of the manuscript that addresses the points raised during the review process.

The three reviewers have provided positive feedback on the paper overall. They recommend acceptance pending some corrections and minor improvements to further enhance the quality of the work. Addressing these suggested revisions will strengthen the paper for final publication.

We look forward to receiving your revised manuscript.

Kind regards,

Shrisha Rao, Ph.D.

Academic Editor

PLOS ONE

Journal Requirements:

Additional Editor Comments:

The authors have done well by addressing the previous comments, and now the reviewers only suggest minor corrections and improvements.

Reviewers' comments:

Reviewer's Responses to Questions

**Comments to the Author**

1. If the authors have adequately addressed your comments raised in a previous round of review and you feel that this manuscript is now acceptable for publication, you may indicate that here to bypass the “Comments to the Author” section, enter your conflict of interest statement in the “Confidential to Editor” section, and submit your "Accept" recommendation.

Reviewer #1: (No Response)

Reviewer #2: (No Response)

Reviewer #3: (No Response)

2. Is the manuscript technically sound, and do the data support the conclusions?

Reviewer #1: Yes

Reviewer #2: Yes

Reviewer #3: Yes

3. Has the statistical analysis been performed appropriately and rigorously? 

Reviewer #1: Yes

Reviewer #2: Yes

Reviewer #3: Yes

4. Have the authors made all data underlying the findings in their manuscript fully available?

Reviewer #1: Yes

Reviewer #2: Yes

Reviewer #3: Yes

5. Is the manuscript presented in an intelligible fashion and written in standard English?

Reviewer #1: Yes

Reviewer #2: Yes

Reviewer #3: Yes

6. Review Comments to the Author

Reviewer #1: This was a well-written, methodologically rigorous, meticulously documented and interesting paper looking at COVID-related circumstances among suicide decedents, using data from CDC’s National Violent Death Reporting System. I’d like to see the authors address the following feedback, however:

1. The introductory epidemiology piece needs some edits for accuracy/clarity. Suicide is currently the 11th leading cause of death, not the 12th. Numbers need to be updated using NVSS 2022 data, and there were over 46,000 deaths by suicide in the U.S. in 2022, not “almost” 46,000. Additionally, the authors often use the term “annually” when they are talking about specific years. This should be edited to refer to specific years.

2. Consider whether reference 4 is really needed – the data it used is the same CDC data referred to in other references.

3. References 11 and 12 refer to the same paper. The authors do not need both.

4. The description of NVDRS narratives in the manuscript is not exactly correct. While the authors frame the narratives around the “salient circumstances” in a victim’s life at the time of death, they are really intended to tell the story of the incident. Per the NVDRS Coding Manual version 6.0:

Narrative accounts of the incident serve multiple purposes:

▪ To briefly summarize the incident (who, what, when, where, why)

▪ To provide supporting information on circumstances that the abstractor has endorsed in an incident

▪ To provide the context for understanding the incident

▪ To record information and additional details that cannot be captured elsewhere

▪ To facilitate data quality control checks for coding key variables

Edits are also needed on line 178, and 463-465 where this description is given again.

5. Line 112 needs to be updated. NVDRS is soon releasing 2022 data. Edit to clarify this and that at the time of the analysis (not “currently) it included over 360,000 suicide and undetermined deaths.

6. Line 273: Minor typo. “Visualized” should be “visualize”

7. The authors should explain where the employed vs. unemployed information came from.

8. Line 336: Appears to be missing a word

9. Line 341: Minor typo. Should be substance use problems, not problem.

10. How were containment measures separated form concerns about containment measures? (Line 348)

11. Line 355 and S11: NVDRS data sharing agreements require narrative excerpts to have details changed or be composites to protect the privacy of decedents. It is not noted in the manuscript or supplemental materials if this was done. This should be done and noted.

12. Line 526: NVDRS is administered by CDC and participating states.

13. Figure 3: It is not clear why there are CIs on this figure given that these are counts.

14. S1: The authors state: “For example, a recent validation study found that NVDRS circumstance variables missed to indicate about a half of cases with known mental health and substance use problems.1 However, the reference called out by the superscript is not given. This is a strong critique of the validity of NVDRS circumstances and needs corroboration.

Thank you for the opportunity to review this work.

Reviewer #2: The present manuscript addresses pandemic-related circumstances among suicide victims during 2020, taking into account a range of pre-pandemic data, sociodemographic and psychosocial factors. The study is relevant given the uncertainty about the relationship between COVID-19 and suicide deaths.

The introduction is well-written, the relevance of the topic is clear, and the line of argumentation is coherent. In the methods section, a detailed description of the data source is provided. Justification for the exclusion of the data collected in 2019 is also sensemaking.

The results are clearly presented, and the supplementary files are comprehensive, providing helpful information about the procedure. Furthermore, the discussion is well-structured, it contains many important and interesting points.

Overall, the manuscipt reads well and I have only identified a few minor issues. These are presented below:

Abstract

• In the last sentence (Conclusion), add „and“ before „can inform“.

Introduction

• Line 86: Add „of“ before „the potential contribution“.

Materials and Methods

• To avoid any misunderstanding, I suggest replacing „COVID“ with „COVID-19“ throughout the methods section and also other sections.

• Line 138: Replace „restrict“ with „restricted“, given that the rest of the methods section is written in past simple tense.

• Line 199: It would be good if you could provide an additional explanation on why being tested positive for COVID-19 was not coded as pandemic-related circumstance. I think that a COVID-19 infection at the time of death is a relevant factor, so I would appreciate if you cound provide a justification for your decision in the revision.

Discussion

• I suggest rephrasing the third sentence as follows: „Victims of suicide deaths with PrC tended to be older and more educated relative to those whose deaths did not include PrC“.

• Line 453: The sentence „These findings can inform future efforts to explore these questions.“ is unclear. Please rephrase it for comprehensibility.

Reviewer #3: This is a very interesting study and the work is well-presented. The authors made substantive changes to address the previous reviewers' comments. This work will add very important contributions to the literature. The authors utilized a novel approach to address their study questions. I recommend acceptance for publication. However, I think this manuscript would be strengthened by MINOR revision early in the manuscript to better describes NVDRS and its data and limitations. This is important because understanding NVDRS is central to contextualizing the analyses and findings. Important clarifications/additions are those that are preceded by ** in the list below and summarized here: NVDRS did not include all states until recently, all regions of participating states may not be included, narrative data is abstracted from CME/LE reports but the information contained within those CME/LE source documents is inconsistent and non-standardized, abstractors have a lot of influence over the narrative data and when reports provide different or conflicting information abstractors record what they believe to be the most accurate narrative.

A few proofreading edits are also noted.

Background-

Lines 77-80: This sentence references “numerous surveys” and “several reports” but only two citations are provided. Modify sentence or include more supporting citations.

Line 86: a word or two is likely missing from this sentence

**Line 95: Re: “these narratives collected information about pandemic-related circumstances” – Might be beneficial to clarify that this information was included in NVDRS only if it was in the CME or LE reports, but that inclusion in those (CME/LE) reports is inconsistent and non-standardized.

Data Source-

**Lines 112-114: This sentence may be a little misleading. Data is not available for all 50 states, DC, and PR dating back to 2003. NVDRS was implemented in 2003 with 5 states participating, and since then an increasing number of states contribute data each year. I don’t believe NVDRS included all 50 states until 2019. Even when a state is represented in the data, the reporting may only include a portion of the counties in the state.

Line 113: (henceforth “suicides”) -I’m interpreting this as meaning for the purpose of this dataset you are referring to both suicides and undetermined manner of death cases as “suicides”. If so, this henceforth statement might be better positioned after your explanation in Lines 120-122 on why you’ve included undetermined deaths.

Line 163: Any vs none – Please clarify that the none are derived from the no/not available/unknown code, which means the “none” category is a mix of cases with none and not available/unknown. I think this is important point because of information you highlight in the S1 appendix: “NVDRS circumstance variables encode valuable information, however these variables are often limited by data availability and structural changes in data processing. For example, a recent validation study found that NVDRS circumstance variables missed to indicate about a half of cases with known mental health and substance use problems.”

Line 170-172: 150-300 words, the case inclusion strategy was previously described to include cases with narratives >= 35 characters, how does this align with the 150 word lower limit? These seem inconsistent. Please clarify this discrepancy.

Measures-

**Qualitative narrative text: Data abstractors have a lot of influence over the narrative data. For example: the NVDRS coding manual (NATIONAL VIOLENT DEATH REPORTING SYSTEM WEB CODING MANUAL VERSION 6.0* (cdc.gov)) provides the following instruction on page 25: “Sometimes information across or within C/ME and LE reports may provide different or conflicting information…If multiple LE reports conflict about the narrative details, record what you believe to be the most accurate narrative. Do the same for multiple C/ME reports.”

Analysis-

Line 273: “visualized” should be “visualize”

Line 301: the second “an” should be “a”

Discussion-

Line 408: the word “did” is likely missing

Please number the tables, figures, and supplemental material sequentially in the order they appear in the manuscript

7. PLOS authors have the option to publish the peer review history of their article (what does this mean?). If published, this will include your full peer review and any attached files.

Reviewer #1: No

Reviewer #2: No

Reviewer #3: No

---

## [Author Response · Author response to Decision Letter 1]

23 Sep 2024

Please see attached response letter.

---

## [Editor Report · Decision Letter 2]

30 Sep 2024

Psychosocial and Pandemic-Related Circumstances of Suicide Deaths in 2020: Evidence from the National Violent Death Reporting System

PONE-D-24-21265R2

Dear Dr. Mezuk,

We’re pleased to inform you that your manuscript has been judged scientifically suitable for publication and will be formally accepted for publication once it meets all outstanding technical requirements.

Kind regards,

Shrisha Rao, Ph.D.

Academic Editor

PLOS ONE
---

## [Editor Report · Acceptance letter]

4 Oct 2024

PONE-D-24-21265R2 

PLOS ONE

Dear Dr. Mezuk, 

I'm pleased to inform you that your manuscript has been deemed suitable for publication in PLOS ONE. Congratulations! Your manuscript is now being handed over to our production team.

Kind regards, 

on behalf of

Dr. Shrisha Rao 

Academic Editor

PLOS ONE